# Reconstruction of four-dimensional rockfall trajectories using remote sensing and rock-based accelerometers and gyroscopes

Andrin Caviezel[1], Sophia E. Demmel[1], Adrian Ringenbach[1], Yves Bühler[1], Guang Lu[1], Marc Christen[1], Claire E. Dinneen[1], Lucie A. Eberhard[1], Daniel von Rickenbach[1], and Perry Bartelt[1]

[1]WSL Institute for Snow and Avalanche Research SLF, CH-7260 Davos Dorf

**Correspondence:** Andrin Caviezel (andrin.caviezel@slf.ch)

**Abstract.** This work focuses on the in-depth reconstruction of the full set of parameters of interest in single block rockfall trajectories. A comprehensive understanding of rockfall trajectories holds the promise to enhance the application of numerical models for engineering hazard analysis. Such knowledge is equally important to investigate wider cascade problems in steep terrain. Here, we present a full four-dimensional trajectory reconstruction of the *Chant Sura EOTA$_{221}$* rockfall experiment. The data analysis allows a complete kinematic description of a rock's trajectory in real terrain and underscores the physical complexity of rock-ground interactions. In-situ accelerometer and gyroscope data are combined with videogrammetric and unmanned aerial systems mapping techniques to understand the role of rock rotations, ground penetration and translational scarring in rockfall motion. The exhaustive trajectory reconstruction provides information over the complete flight path such as translational velocity vectors, angular velocities, impact duration and forces, ballistic jump heights and lengths. The experimental data provides insight into the basic physical processes detailing how rotating rocks of general shape penetrate, rebound and scar ground terrain. The data serves in future as a calibration basement to enhance numerical rockfall modelling.

## 1 Introduction

There is considerable uncertainty in rockfall engineering practice regarding how to predict runout distances, jump heights and lateral dispersion of falling rocks. This information is needed by hazard mitigation experts to develop danger maps and plan cost-efficient protection methods such as rockfall dams and nets. An uncertainty, predominantly arising by the vast and diverging literature adapting the coefficient of restitution (COR) concept to rockfall problems. While some models restrict themselves to a single coefficient, quantifying the energy dissipation as velocity magnitude or kinetic energy loss (Paronuzzi, 1989; Azzoni and Defreitas, 1995; Bozzolo and Pamini, 1986; Chau et al., 1999), the most common description involves both a tangential and a normal restitution coefficient (Evans and Hungr, 1993; Budetta and Santo, 1994; Asteriou and Tsiambaos, 2018). More recent extensions try to incorporate various post-impact dependencies of the block velocity on the pre-impact kinematics (Pfeiffer and Bowen, 1989; Dorren et al., 2004; Chau et al., 2002; Bourrier et al., 2009) and even surface material (Uzi and Levy, 2018). Consequently, various definitions on CORs have been proposed, such as the kinematic, kinetic or energy COR (Asteriou et al., 2012) without converging to a clear consensus. However, processes emerging from altered impact conditions, such as deviations from normal impact configurations, high rotational speeds of the impacting object, etc. remain

challenging and are not unambiguously solvable within any COR framework. These impact configuration define the speed, jump height and dispersion of falling rocks in natural terrain and are the core problem when developing physics-based dynamic models for rockfall hazard mitigation and planning (Leine et al., 2014).

With the advent of affordable computing power, three-dimensional rockfall modelling has become standard technology for risk assessment (Leine et al., 2014; Dorren, 2010; Bourrier et al., 2009; Lan et al., 2007; Agliardi and Crosta, 2003). The application of numerical modelling has the advantage of providing spatially inclusive information on runout distances, velocities, jump heights and impact energies as a function of terrain. When historical data is unavailable to ascertain rock behaviour, numerical simulation becomes the primary tool that engineers can apply to quantify the effectiveness of proposed mitigation measures. A significant problem with numerical approaches, however, is the selection of constitutive parameters governing the rock-ground interaction. This problem is critical because of the wide range of materials and geomorphologies encountered in the rockfall problem. These range from hard bedrock, scree fields, hard frozen mountain soils to highly deformable, soft soil substrates. The problem is compounded by presence of surface vegetation.

Experimental trajectory analysis would serve to calibrate constitutive models and/or numerical model input parameters (Giani et al., 2004) - overcoming the limitation of being solely dependent on case study back calculations. Trajectory analysis aims at the full reconstruction of the 3D flight path in order to gain insights into slope relevant kinematics. A standard approach is to deploy one or more high speed video cameras with a maximized field of view in order to track the majority of the rockfall path (Giani et al., 2004; Dorren et al., 2005; Ushiro et al., 2006; Dorren et al., 2006; Bourrier et al., 2012; Spadari et al., 2012; Giacomini et al., 2012) or to downscale the experiment or sections of it to laboratory size (Cui et al., 2017; Gratchev and Saeidi, 2018; Gao and Meguid, 2018). Usually, the degree of slope coverage is a trade off between field of view and frame rate of a given camera setting, explaining why most presented experimental reconstruction are restricted to specific trajectory sections. Some recent examples for a full-slope determination of single-block rockfall dynamics applied unmanned aerial systems (UAS)-based mapping and trajectory back analysis for field studies (Saroglou et al., 2018) or seismographic and videogrammetric techniques for controlled single-block experiments (Hibert et al., 2017; Saló et al., 2018).

High image resolution videogrammetry can provide the missing data to fully reconstruct rockfall trajectories, especially the ballistic flight path between two impacts. Once the position coordinates from lift-off and impact locations are available, flight parabolas can easily be fitted yielding valuable information on velocities, jump heights and lengths. Derivation of the impact coordinates are extracted via *a posteriori* impact mapping. A second reconstruction pathway is via dense cloud reconstruction (DCR) which is derived from high resolution stereoscopic videogrammetry. With this method it is possible to identify rock position and track trajectories irrespective of the kinematic state of the rock; that is, if the rock is jumping, rolling or sliding.

In this paper, we showcase a novel experimental methodology of combined UAS techniques with in-situ sensor data enabling exhaustive trajectory reconstruction. This approach is unique because we obtain experimental data from two different spatial coordinate systems: a fixed, global coordinate system (UAS) and a moving, local coordinate system (in-situ) moving with the rock. With this approach not only the flight kinematics but also the impact behaviour can be analyzed in great detail. Comparison of high resolution digital elevation models obtained before and after the experiment then allows us to identify the location and dimension of ground scars. Thus, we are able to relate measured accelerations and changes in rotational velocity

to ground deformation and, therefore, the degree of energy dissipation in the ground substrate. The presented work features a combination of remote sensing techniques, low-power microelectromechanical sensor systems and a possible extension of photogrammetric processing work-flows for dynamic rockfall data and is clear evidence, that uniform restitution coefficients are an over-simplified model description for rock-soil interactions.

## 2 Experimental Test Site and Methods

### 2.1 Experimental Site

The experimental site Chant Sura (46.74625 N, 9.96720 E) is located roughly 12 km south-east of Davos, Switzerland (see Fig. 1a). The release point is located at 2380 meters a. s. l. yielding a projected travelling distance of $\sim 250$ meters for the boulders indicated as red dots in Fig. 1. The soil characteristics features typical alpine meadow interspersed with rocks featuring slope angles between 40-80 degrees in the transition zone and a rough scree field runout for the relevant extent in Fig. 1b. The displayed surface ruggedness or vector roughness measure (VRM) is calculated as a 3-dimensional dispersion of the surface normal over a 11x11 neighbourhood of the UAS derived raster digital elevation model of 4 cm resolution (Sappington et al., 2007). A prominent feature is an almost vertical cliff located in the upper part of the slope. The upper level of the cliff and the beginning of the scree field are outlined with orange lines in Fig. 2. It is an ideal representative of a prototypical alpine environment subjected to rockfall hazards. The test site surpasses its predecessor in terms of a larger vertical drop and longer transition zone both favouring higher rock energies. Additionally, no man-made infrastructure or transport route is endangered and accessibility is given via the pass road. Two sets of ground-control points are evenly distributed over the test site each optimally oriented for recognition either via front view videogrammetry or top-view UAS imagery. Their accurate 3D positions are recorded with a high precision differential Stonex S800 GNSS receiver.

### 2.2 Experimental Methods

A high resolution digital surface model (DSM) is generated pre- and post-experimentally via aerial remote sensing using a DJI Phantom 4 Pro equipped with its internal 20 MP camera. Flight planning is achieved with the photogrammetry tool UgCS Pro ensuring precise flight control on steep slopes and sufficient image overlap. Forward overlap was set to 80%, side overlap to 60% respectively. UAS flights were executed at a UAS-ground separation distance of 75 m. An area of 0.2 km$^2$ was covered, 483 photographs taken, yielding a point density of 630 points/m$^2$. The obtained UAS imagery was processed using the (at that time) latest AgiSoft PhotoScan Pro v1.4.3, a commercial software extensively used in the UAS community (Agisoft, retrieved 08.11.2018). For the absolute orientation, recorded ground control points are used. The DEM then can be exported in different resolutions via the PhotoScan interpolation algorithm. This photogrammetric work flow originally introduced for snow depth mapping (Bühler et al., 2012, 2017) works equally well on snow-free terrain and provides a DSM resolution of 5 cm and altitude uncertainties of $\pm 3$ cm .

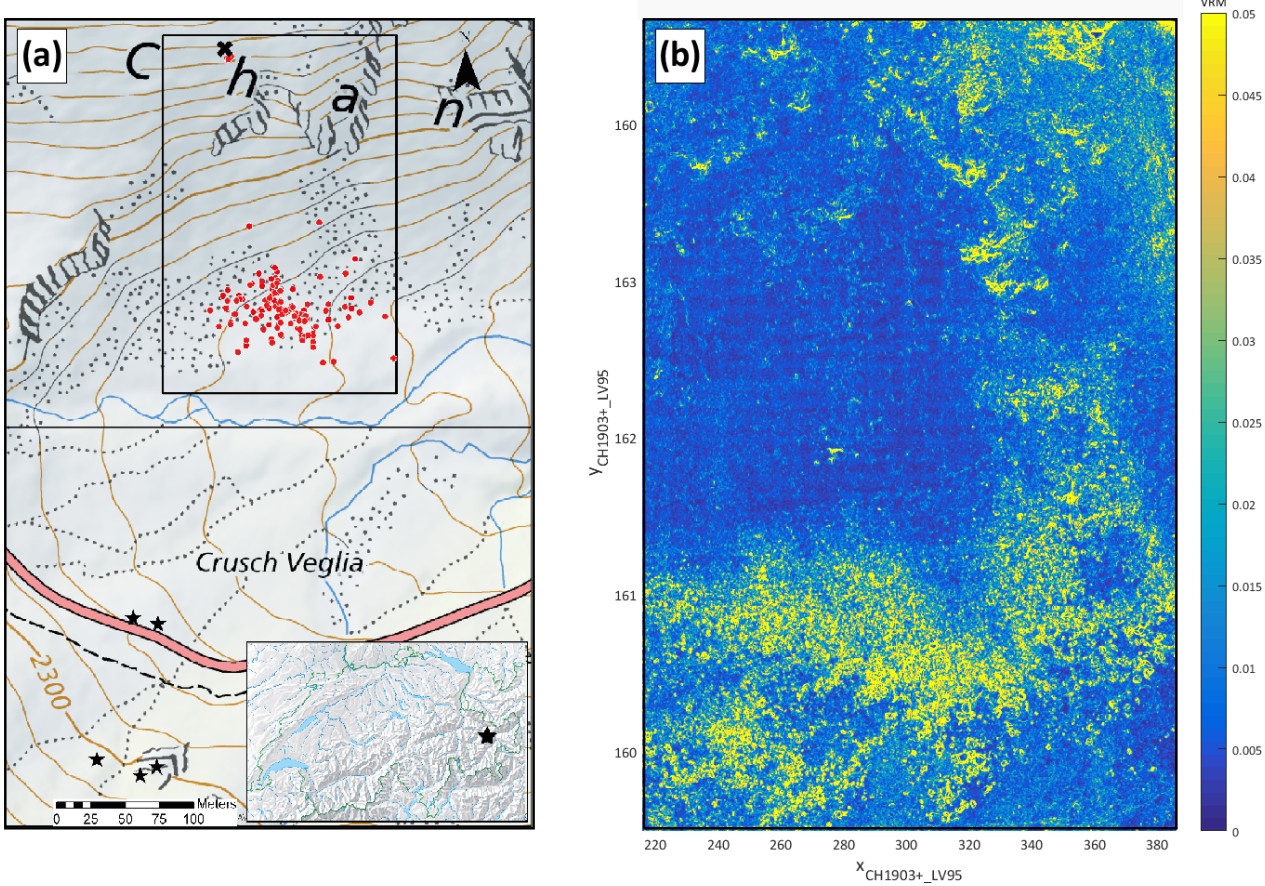

**Figure 1. (a)** Test site overview: Indicated is the release point (black cross), the full deposition data set comprising 125 deposition points (red dots). The camera positions used for the two RED Epic videogrammetry are indicated as black stars. Inset: Geographical location of the *Chant Sura* test site within Switzerland. **(b)** Surface ruggedness - vector roughness mesure (VRM) - for the indicated extend from (a) determined via the surface normal dispersion, highlighting the high ruggedness in the scree field runout (Sappington et al., 2007).

Static stereo-graphic videogrammetry for each rockfall trajectory is performed via two spatially separated RED EPIC-W S35 Helium cameras each equipped with a Canon EF 24-70mm Lf/2.9 (experimental set *RF16*) or Zeiss Otus 55mm/1.4 (*RF18*) lens in order to guarantee optimal image quality. The 8K video footage consists of a 25 frames per second image stream with an image resolution of 8192x4320 pixels. Synchronization of the two cameras is achieved via a Tentacle Sync Lock-it set and an acoustic signal using a traditional clap-board for redundancy. Post-processing of the images includes minimal rendering via Adobe Premiere with respect to image quality and saving each individual frame to JPEG format. The camera positions are indicated as black stars in Fig. 1a, being close to the road for *RF16* and further up the counter-slope for *RF18* owing to the fixed focal length.

The in-situ sensor is a StoneNode v1.1 mounted in the rock's center of mass, recording accelerations up to 400 $g$ and rotations up to 4000 °/s at an acquisition rate of 1 kHz and a recording time of several hours allowing for recording an entire experimental set consisting of 5 to 15 rotations (Caviezel et al., 2018; Niklaus et al., 2017).

The test rock is the platy edition of a perfectly symmetric EOTA (norm rock of the European Organization for Technical Assessment used in standardized rock fence testing procedures in official European Technical Approval Guidelines) made from reinforced concrete with a weight of 780 kg (EOTA$_{221}$, see inset in Fig. 2). The artificial rock ensures full control over rock shape and repetitive experimental series with the same rock shape and weight. The rock is released via a hydraulic platform, its deposition point is measured with a high precision Trimble GeoXH differential hand-held GNSS with an accuracy of 5 cm and transported back to the release platform with an Airbus H125 helicopter.

## 3    Data and Post Processing

This work focuses on the in-depth reconstruction of the full set of parameters of interest in single rockfall trajectories. Thus, we exemplary scrutinize five individual experimental runs belonging to a larger experimental data set consisting of multiple runs with the EOTA$_{221}^{780kg}$ rock. The investigated runs of two experimental days are labelled as *RF16 Run 2,4,5* and *RF18 Run 1,4*. Raw data comprise the GNSS deposition locations, StoneNode v1.1 data streams for each trajectory, RED EPIC video streams, the pre- and post-experimental UAS imagery as well as two series of in-field mapped scars. The aerial overview of the treated data set is given in Figure 2a, showing the UAV derived orthophoto of the experimental site available in a 3 cm resolution. Marked are the release point (X), the projected rockfall trajectory paths (dotted lines). The transition zone is indicated by two orange lines, confined by the upper contour line of the cliff face and the upper scree field boundary. The white squares indicate the mapped scar positions for the two peripheral runs. The final deposition locations for the individual runs are plotted as pink triangles. Figure 2b shows the elevation difference map derived from the pre- and post-experimentally generated digital elevation model in a 5 cm resolution for *RF16*. The difference range is set to ±0.1 m for visibility purposes as the major scarring contributions predominantly occur within this range.

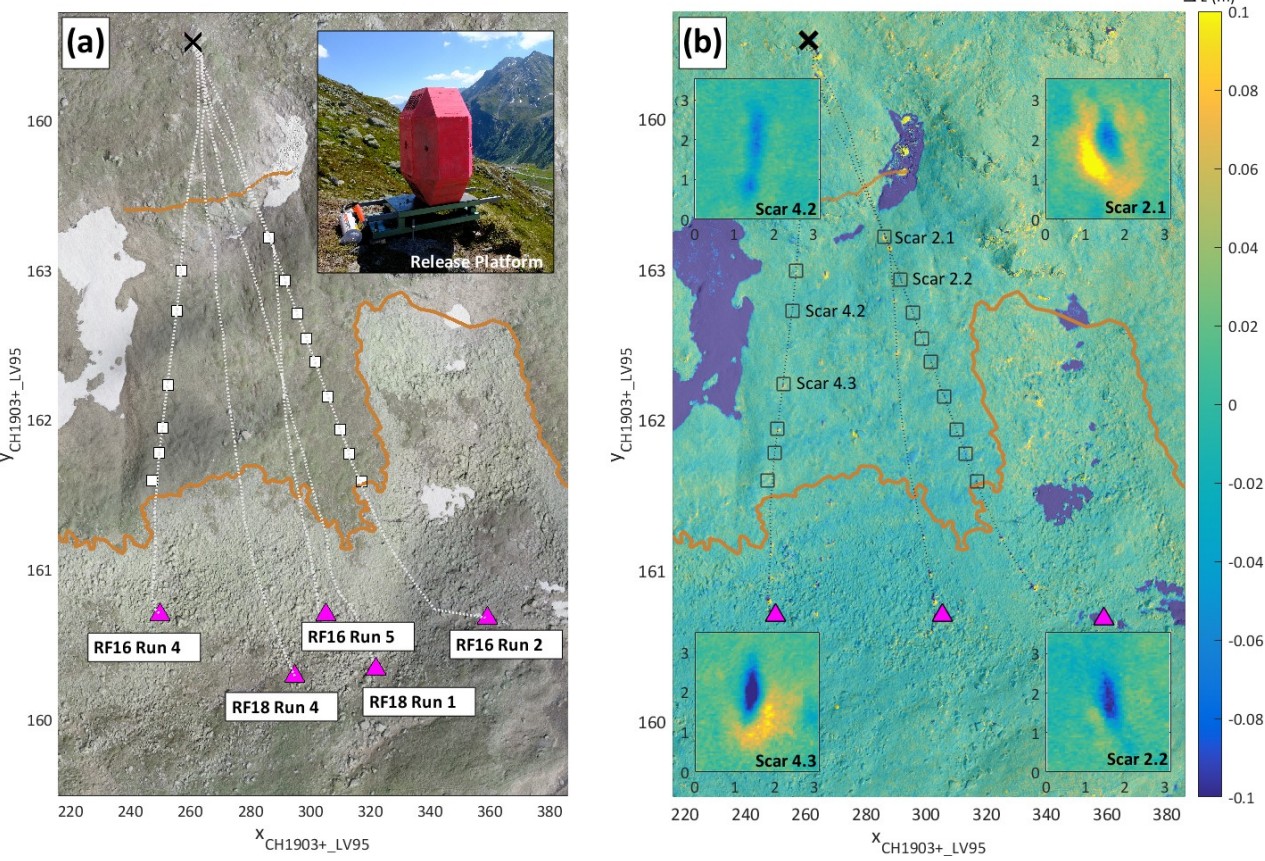

**Figure 2. (a)** UAS derived orthophoto of the experimental test site with marked release point, projected travel path of the investigated trajectories alongside with the mapped scars and the final deposition positions of the rocks. The level of the cliff and transition to the scree field are outlined. The inset shows the release platform with the 780 kg rock in starting position. **(b)** Elevation difference map derived from a pre- and post-experimental UAS generated digital elevation model for *RF16*. The mapped scars for the presented *RF16* runs are indicated with black squares. The course of the trajectories is also visible in the scree field where elevation differences occur due to shifted rocks. Note, that the elevation loss of roughly 10 cm in the snow fields serves also as qualitative validation of the measured differences. Insets: Characteristic scarring pattern without (Scar 2.2 and 4.2) and with material accumulation (Scar 2.1 and 4.3) in resultant travel direction.

## 4  Results

The upper part of the trajectory path is referred to as acceleration/stabilization phase, where the wheel shaped rocks - if not stopped immediately caused by an impact on their flat side - gain momentum and start to stabilize around their largest moment of inertia. This subsequently leads to the wheel like descendant behaviour with angular velocities of 1700-2000°/s (9.4-11.6 rad/s). The stabilization depends on the individual impact conditions and features thus a stochastic nature. All rocks exhibit a stabilized motion when entering the transition zone, starting from the slope cliff. For two peripheral runs, the impact scars have

**Table 1.** Comparison of scar extents, length $L$, width $W$ and depth $D$ from in-field measurements and from the altitude difference map denoted with superscripts *ifm*, and *adm* respectively. Error estimation for both techniques amount to $\pm 3$ cm.

| Scar | $s_L^{ifm}$ | $s_L^{adm}$ | $s_W^{ifm}$ | $s_W^{adm}$ | $s_D^{ifm}$ | $s_D^{adm}$ |
|------|------|------|------|------|------|------|
| 2.0 | – | 1.46 | – | 0.49 | - | 0.08 |
| 2.1 | 2.52 | 2.07 | 0.34 | 0.70 | 0.20 | 0.10 |
| 2.2 | 1.40 | 1.28 | 0.27 | 0.60 | 0.13 | 0.12 |
| 2.3 | 1.80 | 2.68 | 0.34 | 0.40 | 0.15 | 0.08 |
| 2.4 | 1.55 | 1.68 | 0.32 | 0.67 | 0.18 | 0.10 |
| 2.5 | 1.52 | 1.87 | 0.32 | 1.06 | 0.21 | 0.15 |
| 2.6 | 1.40 | 1.68 | 0.34 | 0.52 | 0.16 | 0.08 |
| 2.7 | 0.82 | 2.95 | 0.36 | 1.25 | 0.27 | 0.11 |
| 2.8 | 0.65 | 1.33 | 0.40 | 1.81 | 0.14 | 0.14 |
| 4.1 | 1.60 | 0.84 | 0.30 | 0.20 | 0.20 | 0.05 |
| 4.2 | 2.27 | 2.55 | 0.27 | 0.45 | 0.22 | 0.08 |
| 4.3 | 2.18 | 1.81 | 0.50 | 0.54 | 0.29 | 0.16 |
| 4.4 | 1.75 | 1.80 | 0.35 | 0.66 | 0.27 | 0.12 |
| 4.5 | 1.50 | 0.85 | 0.20 | 0.60 | 0.21 | 0.08 |
| 4.6 | 1.40 | 1.00 | 0.25 | 0.21 | 0 | 0.04 |

been mapped. The identification of the scars is facilitated in the peripheral transition zone as fewer rocks take this course. After the transition zone, the rocks enter the runout zone composed of a sightly declining rough scree field (see Fig. 1b) yielding to decelerating motion.

## 4.1 Scar Mapping

The elevation difference map in Figure 2b shows clearly discernible scars of the two peripheral runs. The mapped scars are indicated with black squares. The course of the trajectories remains traceable in the scree field where elevation differences occur due to shifted rocks. This information allows together with the temporal information from either the video or sensor stream for a back analysis of the trajectory kinematics within the scarring and assists in the full 3D trajectory reconstruction. The in-field scar mapping includes a GNSS location and a manual measurement of length, width and depth with a measurement

error of 5 cm. UAS scar measures were obtained by taking the $(x,y)$ extent exceeding the measurement uncertainty $\Delta z > 3$ cm and the maximum depth within the scar extent. Table 1 summarizes the mapped values arising from in-field and UAS mapping. The insets of Figure 2b highlight two characteristic scarring patterns: One being a plain convex scarring (*Scar 4.2* and *Scar 2.2*) representative of a splashing, non-accumulating type of scar behaviour. The second type is a mixed convex/concave scarring, that is a convex scar with an additional subsequent material accumulation in resultant travel direction building up a launch pad

for the rock (*Scar 4.3* and *Scar 2.1*). The examined scarring instances are labelled in Figure 3 and in the corresponding sensor

stream insets. The plain convex scarring has little effect on the angular acceleration where the gain and losses are predominantly correlating with slope angle. The mixed scarring, on the other hand, yields to a pronounced decrease in angular velocity. It is governed by soil compaction within the scarring layer, its accumulating to a launch pad facing opposite to the main travelling direction and therefore imposing a rotational drag force.

## 4.2 3D Trajectory Reconstruction

While a post-event UAS back analysis based on scar patterns similarly to Saroglou et al. (2018) leads to highly valuable insights in possible trajectory paths they still miss the temporal information to pin down the exact flight parabola. Because only complete trajectory information, especially jump heights and lengths allow for accurate and thus cost-efficient design and placement of mitigation measures. Here, time information can easily be gathered from the sensor stream or the videogrammetry. Where a scar track is available the impact coordinates are inherently present. If no scar mapping is available, exact impact and lift-off coordinates have to be evaluated via RED imagery, either by manual determination of impact and lift-off positions in the video stream or via dense point cloud reconstruction of the stereoscopic image pairs.

### 4.2.1 A posteriori Impact Mapping

The reconstruction of each flight parabola can be achieved if start and endpoints - as for example given by a scar pair - together with the time interval needed to conquer this given distance are known. Thus, an a posteriori impact mapping (AIM) requires identifying the $(x, y, z)$ coordinates for all impact and lift-off points with corresponding time intervals extracted from the sensor. The video serves as visual identifier between geographic information system (GIS) mapping environment and sensor stream. The use of the equation of motion for each oblique throw then yields the full kinematic information for each trajectory section, that is velocity information, impact and launch angle and consequently the jump heights.

### 4.2.2 Dense Cloud Reconstruction Method

Ideally, a videogrammetric trajectory reconstruction should require no manual input. Here, we show a possible pathway to automatic reconstruction from stereoscopic imagery via dense cloud reconstruction (DCR). Photogrammetric processing of digital images and generating three-dimensional spatial data has become standard for static applications (see Luhmann (2018); Linder (2016); Albertz and Wiggenhagen (2009); Kraus (2007) and references therein). Commercial photogrammetry software is highly efficient and specialized when it comes to the generation of a dense point cloud of a static scenery recorded with a huge number of single images, analogous to the DSM and orthophoto reconstruction.

The application of stereoscopic videogrammetry to moving targets and subsequent automatic target recognition alters the premises significantly. The first key requirement is a set of cameras being able to synchronously trigger with a sufficient temporal rate with respect to the motion under investigation. It becomes obvious that for rocks travelling at speeds of 30 m/s a frame rate of 10 frames/sec (fps) is rather low especially for resolving the runout behaviour often featuring high velocities and rather short and flat jumps. While most available cameras offer 25-30 fps they fail to comply with the second key requirement:

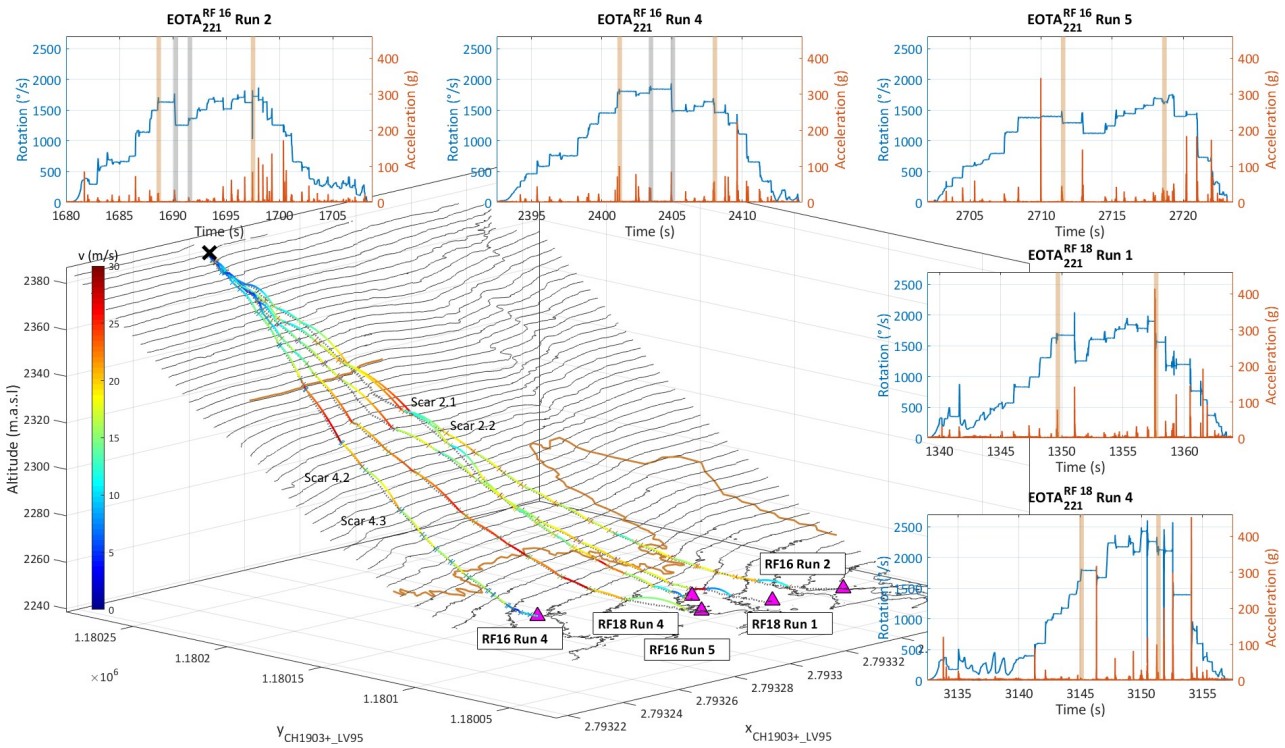

**Figure 3.** Four-dimensional trajectory reconstruction of five selected experimental runs. The trajectory is color coded based on its translational resultant velocity, where the top speeds of roughly 30 m/s are usually reached after the longest airborne phase at the cliff jump. The level of the cliff and the transition to the scree field are outlined. The insets show the according in-situ sensor streams featuring resultant impact accelerations and angular velocities. The sections corresponding to the start of the cliff jump and the entrance into the scree field are shaded in orange in the sensor data plots. For *RF16 Run 2* and *RF16 Run 4*, the investigated scars are shaded gray.

sufficient image resolution when covering a large slope. This is overcome by use of the RED EPIC-W S35 8K camera. The 8192x4320 pixels image pairs allow for sufficient pixel resolution of the rock over the entire slope.

The image feed of *RF18 Run 1* consisting of 492 image pairs is processed through the Agisoft work flow (Agisoft, retrieved 08.11.2018). After image import, alignment of each of those image pairs with highest accuracy and tie and key points limit 5 (using 8000/80000) is performed. Import of the GNSS coordinates of the ground control points are used to align the internal coordinate system to the Swiss Coordinate system CH1903+_LV95.

The next steps are the generation of the sparse point-clouds (setting: *highest accuracy*), optimizing camera alignment, followed by building a dense point-cloud with ULTRA HIGH quality setting. These steps are performed for every image pair individually, consequently delivering 492 dense point clouds. In order to identify the points matching the rock surface in each 10 time step, a surface color based filtering is applied. Due to this procedure, other reddish areas introduce noise. To eliminate this noise in the further examined point clouds, (i) a denoise-function is used (Rusu et al., 2008) followed by (ii) a convex hull

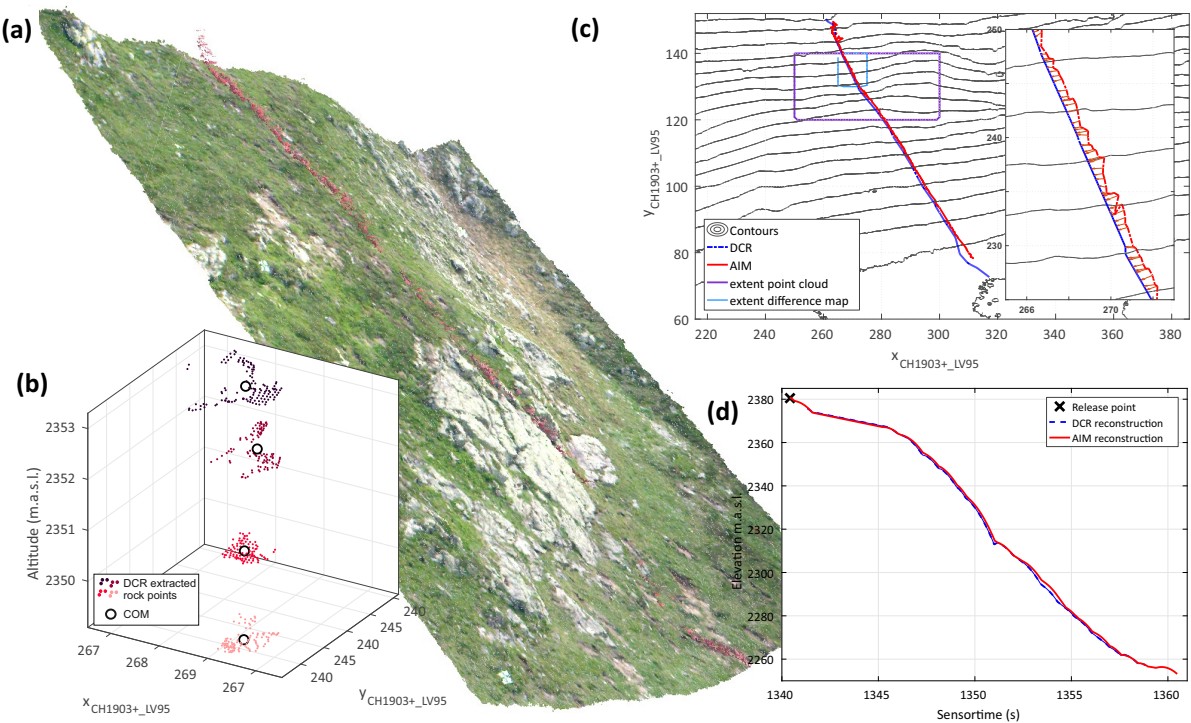

**Figure 4.** Photogrammetric work-flow for the dense cloud reconstruction method. Inset **(a)** depicts a visualization of superimposed reconstructed point clouds of one upper trajectory sector. The pink rock is well distinguishable in most of the image pairs. **(b)** shows the center of mass extraction (o) for matched points clouds after color filtered extraction of the rock at four subsequent positions. **(c)** $(xy)$ planar top view of the reconstructed trajectories of *RF18 Run 1* from both the DCR and AIM (violet frame = outline of **(a)**, blue frame = outline of inset within **(c)**. **(d)** displays the comparison of reconstructed flight parabolas from AIM with fitted parabolas from DCR, showing the $z$ view only.

volume threshold of 1 m$^3$ around the estimated rock position. For the further comparisons with the AIM, the center of mass is extracted with the K-means clustering method. Finally (iii) a logical filter is applied, where a steady downhill movement between 10 subsequently frames of the rock is assumed and any outliers are ignored.

### 4.2.3 Reconstructed Trajectories

5 Figure 3 shows the reconstructed trajectories for the five presented runs along with the corresponding sensor stream displayed as insets featuring resultant impact accelerations and angular velocities. The sections corresponding to the start of the cliff jump and the entrance into the scree field are shaded in orange. For *Run 2* and *4*, the investigated scars are marked in gray. The flight path is velocity color coded, indicating maximal velocities of roughly 100 km/h usually reached after the longest airborne free fall phase above the cliff. Table 2 displays a representative excerpt for selected trajectory sections of *Run 4* and *Run 2*. Denoted

10 are four first flight parabolas in the transition zone beginning with the cliff jump. Displayed are the parameters jump length ($J_L$), jump height ($J_H$), total translational velocities at the flight parabola beginning (i.e. lift-off, $v_{res}^b$) and parabola end (i.e

impact, $v_{tot}^e$) and the correspondingly labelled total angular velocity $\omega_{res}^{b/e}$, kinetic energy $E_{kin}^{b/e}$ and rotational energy $E_{rot}^{b/e}$. The maximal jump heights are derived as the maximum distance between rock center of mass and terrain surface during the flight phases. Moreover, the kinetic, $E_{kin}^{b^{n+1}}/E_{kin}^{e^n}$, as well as the rotational energy transition, $E_{rot}^{b^{n+1}}/E_{rot}^{e^n}$, for the individual impacts is presented. The transition from jump $2_1 \rightarrow 2_2$ corresponds to *Scar 2.1*, etc.. Note that $E_{kin}^{b^{n+1}}/E_{kin}^{e^n} = 1$ would correspond to a perfectly elastic rebound behaviour. Corresponding impact forces are available through the sensor stream. The rotational energy is derived from the gyroscope data such that the ratio between translational and rotational energy can be deduced. The data allows scrutinizing rock-ground interactions, quantifying the basic physical processes of how rocks of penetrate, rebound and scar ground terrain.

Figure 4 shows an overview of the DCR work flow. The reconstructed point-clouds for a few selected flight parabolas from *RF18 Run 1* are depicted in Fig. 4a. The center of mass extraction for reconstructed dense point clouds separated by 0.1 s are displayed in Fig. 4b. Figure 4c features the planar $(xy)$ top view of the reconstructed trajectory where the jitter of the $(x, y)$ position becomes apparent. Fig. 4d shows a well matching comparison of the reconstructed AIM flight parabolas alongside with the parabola fit for the DCR $z$ coordinate.

## 5   Discussion

Trajectory reconstruction becomes a feasible task if high quality scar maps or high resolution imagery is available. For an unambiguous trajectory reconstruction, the temporal dimension has to be known. The classical impact mapping reconstruction methodology yields good results with the drawback of labour intensive manual impact detection and the respective individual judgment on rolling and bouncing behavior. We demonstrated the feasibility of dense cloud reconstruction method, possibly eliminating post-experimental manual input for impact and lift-off detection. A fully computer-aided tracking methodology fuses the demands of automated target recognition for projectiles with continuous motion paths and tracking of a rather erratic behavior via computer-vision reconstruction techniques (Schachter, 2017; Park et al., 2015). Feasibility for degraded contrast between rock and background as well as a fusion with the sensor stream in order to fully automate trajectory reconstruction needs to be elaborated. A promising approach might be the background subtraction of the static point-cloud and thus difference pixel tracking (Benezeth et al., 2010; Makris and Ellis, 2002; Cheng and Kehtarnavaz, 2000). The full set of parameters of interest can be reconstructed, yielding an unprecedented data set on real-scale rockfall experiments. This invaluable information can now be used for calibration purposes of numerical rockfall models, matching simulation performance to experimental results.

Energy dissipation during impacts can be derived from end and start conditions of consecutive parabolas. Usually, energy dissipation during impacts leads to lower lift-off velocities compared to the impact velocities of the preceding impact represented by most kinetic energy ratios $E_{kin}^{b^{n+1}}/E_{kin}^{e^n}$ lying well below 1 in Table 2.. Opposite behaviour is found at the impact between jump $2_2$ and jump $2_3$ where a velocity increase of $\Delta v_{res} = 1.1$ m/s is observed. This is a rather rare example of an almost fully elastic impact behaviour where the entire potential energy intake is converted to the kinetic energy reservoirs. This is reflected in the kinetic energy ratio $E_{kin}^{b^3}/E_{kin}^{e^2} = 1.11$ (see Table 2), even surpassing a fully elastic restitution coefficient of

**Table 2.** Rockfall Trajectory Parameters Of Interest. Denoted are run number and the four jumps $J_{Nr}$ beginning with the cliff jump. Included are jump length $J_L$, jump height $J_H$, translational velocities $v_{res}$, angular velocities $\omega_{res}$, kinetic $E_{kin}$ and rotational energies $E_{rot}$ at the lift-off and impact conditions denoted with superscripts $b$ and $e$ respectively. The last two columns summarize kinetic, $E_{kin}^{b^{n+1}}/E_{kin}^{e^n}$, as well as the rotational energy transition, $E_{rot}^{b^{n+1}}/E_{rot}^{e^n}$, for the individual impacts. The transition from jump $2_1 \to 2_2$ corresponds to *Scar 2.1*, etc.. Note that $E_{kin}^{b^{n+1}}/E_{kin}^{e^n} = 1$ corresponds to a perfectly elastic rebound behaviour.

| Run$_{J_{Nr}}$ | $J_L$ (m) | $J_H$ (m) | $v_{res}^{b/e}$ (m/s) | $\omega_{res}^{b/e}$ (deg/s) | $E_{kin}^{b/e}$ (kJ) | $E_{rot}^{b/e}$ (kJ) | $E_{kin}^{b^{n+1}}/E_{kin}^{e^n}$ | $E_{rot}^{b^{n+1}}/E_{rot}^{e^n}$ |
|---|---|---|---|---|---|---|---|---|
| $2_1$ | 28.14 | 5.65 | 15.75/25.00 | 1627/1632 | 97.3/244.6 | 32.6/31.8 | | |
| $2_1 \to 2_2$ | | | | | | | 0.21 | 0.61 |
| $2_2$ | 15.53 | 1.53 | 11.74/16.74 | 1250/1258 | 53.8/109.3 | 19.3/19.7 | | |
| $2_2 \to 2_3$ | | | | | | | 1.11 | 1.15 |
| $2_3$ | 12.44 | 0.24 | 17.80/21.41 | 1355/1373 | 120.8/176.0 | 22.8/23.4 | | |
| $2_3 \to 2_4$ | | | | | | | 0.55 | 1.18 |
| $2_4$ | 10.25 | 0.58 | 15.77/19.28 | 1495/1508 | 97.2/145.2 | 27.8/28.7 | | |
| $2_1^{mse}$ | 28.21 | 5.67 | 15.80/25.04 | 1627/1365 | 97.7/244.6 | 32.6/31.8 | | |
| $2_1^{mse} \to 2_2^{mse}$ | | | | | | | 0.20 | 0.61 |
| $2_2^{mse}$ | 14.70 | 1.46 | 11.07/16.09 | 1250/1256 | 47.8/101.0 | 19.3/19.7 | | |
| $2_2^{mse} \to 2_3^{mse}$ | | | | | | | 1.04 | 1.16 |
| $2_3^{mse}$ | 11.56 | 0.20 | 16.43/20.07 | 1358/1359 | 105.3/157.1 | 22.8/23.4 | | |
| $2_1^{scp}$ | 29.07 | 5.72 | 16.39/25.62 | 1627/1632 | 104.7/256.0 | 32.6/31.8 | | |
| $2_1^{scp} \to 2_2^{scp}$ | | | | | | | 0.26 | 0.61 |
| $2_2^{scp}$ | 17.19 | 1.50 | 13.03/18.10 | 1250/1258 | 66.2/127.8 | 19.3/19.7 | | |
| $2_2^{scp} \to 2_3^{scp}$ | | | | | | | 1.34 | 1.16 |
| $2_3^{scp}$ | 14.50 | 0.25 | 21.01/24.60 | 1358/1373 | 172.2/236.0 | 22.8/23.4 | | |
| $4_1$ | 28.1 | 2.1 | 21.6/28.8 | 1808/1804 | 181.9/323.4 | 40.6/40.9 | | |
| $4_1 \to 4_2$ | | | | | | | 0.42 | 0.96 |
| $4_2$ | 14.5 | 0.5 | 18.6/22.2 | 1769/1777 | 135.1/192.2 | 39.3/39.8 | | |
| $4_2 \to 4_3$ | | | | | | | 0.50 | 1.08 |
| $4_3$ | 25.2 | 2.8 | 15.6/22.0 | 1843/1843 | 95.3/188.2 | 43.0/43.1 | | |
| $4_3 \to 4_4$ | | | | | | | 0.47 | 0.65 |
| $4_4$ | 14.9 | 1.2 | 15.1/19.1 | 1490/1495 | 89.3/143.0 | 28.2/28.6 | | |

1.0. Detailed examination for the transition $2_2 \rightarrow 2_3$ yields an altitude change of $\Delta h = 1.11$ m leading to a potential energy difference of 8.49 kJ. The calculated energy intake, however, amounts to 17.3 kJ leading to a energy gap of 8.82 kJ, demanding for an additional 1.15 m of altitude change.

In order to elaborate this mismatch and the variability of reconstructed parameters, we post-processed transitions $2_1 \rightarrow 2_2 \rightarrow 2_3$ additionally with a *maximum scar extent* (mse) and a *single contact point* (scp) approach. The former approach sets the scar length to the maximal extent where difference pixels of $\Delta z > 3$ cm in the altitude difference map are discernible - as opposed to the extent of a coherent area with $\Delta z > 3$ cm. The latter sets the impact point to the middle of the scar, being equivalent to a restitution coefficient based rebound model. Where available the scar midpoints are set as the in-field recorded GNSS coordinates. The according values are shown in Table 2 with the superscripts $2_{J_N r}^{mse/scp}$, respectively.

The maximal scar extent treatment reduces the velocity increase in the transition $2_2 \rightarrow 2_3$ to $\Delta v_{res} = 0.34$ m/s at an increased altitude change between impact and lift-off of the $\Delta h = 1.82$ m. The energy intake results in 7.4 kJ corresponding to an altitude change of 0.96 m. The energy gap thus is closed and a rather realistic impact behaviour with small energy dissipation is matched. The single impact point treatment, on the other hand, increases the velocity jump for $2_2 \rightarrow 2_3$ to $\Delta v_{res} = 2.91$ m/s with zero altitude change between impact and lift-off. The energy intake thus amounts to 47.4 kJ corresponding to an altitude change of 6.2 m leading to a heavy mismatch with respect to the experimental trajectory. Major uncertainties for translational variables are introduced during the impact/lift-off position placement. Thus the presented reconstructed velocities in Table 2 succumb to an uncertainty of $\pm 0.5$ m/s. Jump heights remain rather unaffected with the variation of roughly $\pm 0.1$ m, owing to the fact that the temporal uncertainty is small and does not allow for largely altered projectile motion. The measurement precision of the gyroscope, finally, is extremely precise, yielding a maximal jitter of only $\pm 5°$/s.

It becomes obvious that scarring mechanisms are crucial for correct energy treatments. While scarring normally leads to a reduction in translational velocity, the change in rotational speeds is rather small in the transition zone. Interestingly, a significant reduction in rotational speed is distinguishable for the mixed convex/concave scarring pattern. A comparison of pre- and post-impact rotational energy shows a reduction of rotational energy to 61% in *Scar 2.1* and 65% in *Scar 4.3* (see Table 2), while purely convex scarring leaves the rotational speed fairly unaltered, even increase in rotational speed has been observed. This confirms the complex rock-ground interactions during the short impact times as presented by Caviezel and Gerber (2018). Future work will include a comprehensive screening of the trajectory parameters as well as detailed investigation of scarring effects and surface irregularities Gratchev and Saeidi (2018). A comparison with energy considerations derived from seismic analysis might be of interest (Vilajosana et al., 2008; Hibert et al., 2017; Saló et al., 2018).

The presented approach is focusing primarily in gathering extensive data both to enhance the process understanding and consequently for model calibration purposes. Up-scaling of certain experimental techniques for monitoring applications could be envisioned. The in-situ sensors for example could be programmed as low-power monitoring devices, starting its measurement upon a triggering signal such as for example a threshold rotation. Videogrammetric techniques always lack bad weather and low visibility suitability and thus are more suited for self-contained experimental setups. Shifting the automated target tracking to lidar/radar based devices might open up new opportunities for continuous surveillance with subsequent trajectory

reconstruction in an event case. Future work will include adaption of the presented approach to multiple test sites and a possible adaptions to labor-scale experimental setups in order to overcome logistic limitations when studying stochastic processes.

## 6 Conclusions

In this paper we have used a combination of remote sensing techniques and in-situ sensor measurements to reconstruct four-dimensional rockfall trajectories in real terrain. Using this approach we obtain complete data of the parameters of interest: jump heights and jump lengths, rock spin, and the change in acceleration at the point of impact. Such an exhaustive data set facilitates the calibration of numerical rockfall models, independent of their implementation method. Additional information on scarring duration, extent and depth allow us to identify energy dissipation mechanisms for soil substrates. This is a long-standing problem in rockfall engineering.

The preliminary analysis of the data already has generated results of practical interest. A general characteristic of the experimental trajectories is the over-riding presence of flat jumps. Jump height is a crucial parameter in rockfall engineering, especially for design and placement of mitigation measures. Overestimation of jump heights leads to higher mitigation expenses. Flat jump heights appear to result from a complex interaction of rock geometry, surface roughness, rock spin and soil scarring. An in-depth analysis of the full data set comprising more than 50 fully reconstructed trajectories for different masses and shapes, will be needed in order to disentangle this sophisticated interplay.

One important conclusion from our measurements is that it is not possible to describe the complicated rock-soil interaction process with uniform restitution coefficients alone. Restitution coefficients describe the relationship between the incoming and outgoing velocities but provide no information concerning the decelerating forces at impact. These forces depend on the impact configuration and therefore the orientation of the rock. They act over short time periods and are impossible to average or linearize. Without a methodology to consider rock geometry, surface roughness, and soil scarring, the rock-soil interaction process is overly simplified and cannot be effectively used to make consistent runout or jump altitude forecasts.

Reconstructing rockfall trajectories is therefore key in establishing the relationship between geological and geomorphological setting to rockfall runout and dispersion. Preferably, the experimental methodology is expanded to different locations with a wider set of grain sizes. Quantifying the rebound behaviour spanning several mesoscale roughness levels is of fundamental interest to cover a wider range of naturally existing terrain classes subjected to rockfall hazards. Additionally, adding structures such as rockfall nets and/or dams would result in unprecedented examination of mitigation measures under realistic conditions with a fully determined incoming projectile - as opposed to existing, artificial vertical drop test setups.

*Data availability.* The codes and the data used in this study are accessible upon request by contacting A. Caviezel (caviezel@slf.ch). The data set will be publicly available once full analysis has been concluded.

*Author contributions.* A.C., Y.B., P.B. and M.C conceived the experiment. All authors contributed in the data acquisition in the *Chant Sura* experiment. A.C and S.E.D performed the AIM trajectory reconstruction and trajectory analysis, A.R. and A.C. the DCR reconstruction. UAS data acquisition and post-processing were performed by Y.B, D.v.R., and L.A.E.. A.C. and P.B. wrote the manuscript with discussions and improvements from all authors.

5   *Competing interests.* The authors declare that they have no conflict of interest.

*Acknowledgements.* We thank the municipality of Zernez, Switzerland, and Emil Müller for the permission to conduct experiments on the *Chant Sura* site. Special thanks go to Helibernina for their repetitive precision slinging. We thank Matthias Paintner for his support and guidance with the RED Epic handling and the 8K videogrammetry editing.

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
