# Peer review of "Reconstruction of four-dimensional rockfall trajectories using remote sensing and rock-based accelerometers and gyroscopes"

_Earth Surface Dynamics, 2018_

## Referee Comment (RC1) · Anonymous Referee #1 · 7 Nov 2018

General comments:

Caviezel et al. present the results of a series of empirical rock fall experiments, and present a new method for reconstructing 4D rock fall trajectories using a combination of photogrammetry and videogrammetry. The research is evidently well-planned and executed (and, as an aside, sounds like a lot of fun!). The paper is generally well-written, methods are (mostly) comprehensively described, the figures are of high quality, and the discussion is coherent. I recommend its acceptance subject to minor revisions.

As well as addressing my comments below, the authors should consider expanding the discussion to include consideration of the wider implications of the work. The work

essentially showcases the results of a largely self-contained, manageable field experiment. But how well might the approach scale? Is it feasible, or might it be in the future, to upscale this to rockfall monitoring for an entire mountainside, and over an extended period of time, for example?

The paper is also light on references - the Discussion, for example, only contains a single citation. The authors should revisit this section, and discuss their findings in the context of the current literature and include additional references as appropriate.

Line-specific comments:

Title - I'd change 'three-dimensional' to 'four-dimensional', since the temporal aspect of the rockfall reconstruction is something that you have quantified, and spend a large portion of the paper describing and discussing.

Abstract – first line – not just useful for engineering hazard analysis, also valuable for investigating wider sediment cascade problems in steep terrain. The paper in general is written with the focus on rockfall hazard assessment, which is perfectly valid, but the implications of the work go further. The authors should acknowledge this, and spend a little time elaborating on the wider landscape evolution context in which the research sits - perhaps in the discussion.

P3, L6 – what were the forward and side overlaps for the UAV photography? What was the mean flying height or UAV-ground separation distance?

Discrepancy between detail provided for videogrammetric analysis (next para), and DSM derivation – need to expand former. How many photographs, what photogrammetry software, etc etc.

P4, L19 – this is a key sentence for the paper as a whole and should come much earlier, ideally in the introduction. It would even be an appropriate sentence to begin the entire paper with.

P7, L2 – OK, so WHY would someone want/need to reconstruct the flight parabola

component of the rockfall trajectory? Expand this sentence a little to make this clear. You make reference to the use of jump heights for rockfall engineering in the Conclusion, but I'd suggest introducing this here.

Figure 3 – very nice, I like this a lot!

P7, L21 – sentence requires supporting references.

P8, L1 – for consistency, this is the level of detail you should go into earlier with regard to photogrammetric derivation of the slope topography. Please also include a reference to the precise version of PhotoScan that you used – perhaps place this reference earlier.

Also, were all photographs used to build a single, dense 3D point cloud, or were multiple point clouds generated (e.g. one per run)? It seems as though you only generate a single cloud, but this needs clarifying in the text. I'm a little confused on this part of the workflow, and readers may be too. It's impressive stuff, though, given the data volume.

Technical corrections:

- No technical corrections identified, aside from some careful proof-reading to improve the written English in a handful of places.

---

## Referee Comment (RC2) · Anonymous Referee #2 · 19 Nov 2018

This contribution presents a full-scale experiment of rock fall trajectography coupling advanced techniques such as 8k-videogrammetry, uav and a 780kg block equipped of accelerometers. This is a state-of-the-art experiment, with a heavy logistic, and most of the paper is of course devoted to describe this experiment (sensors and procedures). The implications of observations made during this experiment are only shortly discussed (scaring effects for example), and the overall experiment includes actually only five runs with a particular shaped block. But the experiment described in this contribution is innovative and I am sure it will be useful for future research in experimental trajectography. The paper is well written and I recommend accepting it with minor corrections.

[Figure]

General comments:

1. Rockfall trajectography is one of the most popular topics in the geohazards community, with numerous publications. Very few references are actually given in this paper. I would suggest the authors to introduce shortly how their experiment compares with previous experiments (not only those of the authors' group).

2. I would appreciate to have a figure with a picture of one of the impact point and the corresponding point cloud after impact. That would help to understand the quality of the data and the type of information available.

3. I don't think that the few observations made in this experiment (only 5 runs with a wheel-like block) can be generalized to support the conclusions on jump height and mitigation measures dimensioning (p12, l8). Some extra cautions have to be taken before jumping to these conclusions

4. It's a Rolls-Royce experiment, involving a heavy logistics (even a helicopter). Such experiments are definitively useful, but the number of runs is naturally low and may be a limitation when studying stochastic processes. Do you think it would be possible to downscale this kind of experiment, keeping the same monitoring techniques?

Typos:

P1 l19: selection of constitutive

P2 l3: between field of view

---

## Author Comment (AC1) · 19 Nov 2018

We thank the referee for his positive conception of our submission and his suggestions for improvements.

The manuscript has been amended to incorporate the referee's suggestions in the following way (and can be reviewed in the supplemented file):

- Title: We changed it to "four-dimensional" as the temporal dimension is of significant interest in the shown reconstruction, indeed.

- Abstract: We enlarged the significance statement, as the referee points out correctly,

that a complete understanding of rockfall kinematics is a essential part of general cascading effects in gravitational sediment transport.

- The UAS methodology has been supplemented with addtitional technical information for the interested reader, such that the used settings and workflow is reconstructable.

- We ammended all the minor unclarities (P3,L6; P4,L19; P7,L2;P7,L21) with short additional explanations and additional references.

Additional changes: Figure 1 was ammended, such that colorbar ticks are fully readable and the location of the test site is placed more accurately in the "Switzerland" inset. Any small inconsistencies or unclarities in terms of the DCR work-flow should be phrased more clearly. Figure 4 has also been improved: Caption are ammended, readability and self-explanatory style of figure improved.

On behalf of all of the authors, yours sincerely,

Andrin Caviezel

Please also note the supplement to this comment:
https://www.earth-surf-dynam-discuss.net/esurf-2018-74/esurf-2018-74-AC1-supplement.pdf

**Supplement:**

[revised manuscript text omitted]

---

## Author Comment (AC2) · 21 Nov 2018

Dear referee,

We thank for the positive conception of our submission and your suggestions for improvements. We deliberately focused on the experimental methodology and its implementation in order to write a compact publication. An in-detail analysis of the experimental results and their possible implications to constitutive models and calibration routines is on-going work.

Please find below the response on your remaining criticisms:

[Figure]

*1. Rockfall trajectography is one of the most popular topics in the geohazards community, with numerous publications. Very few references are actually given in this paper. I would suggest the authors to introduce shortly how their experiment compares with previous experiments (not only those of the authors' group).*

In order blend our work into the past and ongoing research of rockfall trajectography, we extended the introductory part (Introduction, paragraph 2) by numerous references, focusing especially on published experimental work. As already adked for in RC1, the overall manuscript has been extended with more references to relevant work. Slight rephrasing of the given paragraph, emphasizes the distinction of latest research (Saroglou et al., 2018, Ushiro et al., 2006; Hibert et al., 2017; Salo et al., 2018, our work) on an experimental reconstruction methodology aiming for a full-slope trajectographic reconstruction. The revised introductory paragraph and phrasings are marked in blue in the attachment.

*2. I would appreciate to have a figure with a picture of one of the impact point and the corresponding point cloud after impact. That would help to understand the quality of the data and the type of information available.*

We ammended Figure 4, especially panel (b) and the overal caption such that the quality and type of data of the DCR reconstruction workflow is more easily understandable. Panel (a) depicts a visualization of superimposed reconstructed point clouds of one upper trajectory sector. The pink rock is well distinguishable in most of the image pairs. That is, this figure is a direct visualization of data quality. Inset (b) has been redesigned and labelled such that the extracted point clouds of the rock is emphasised. It shows the residual points of the rock and its centre of mass extraction after colour-filtered extraction of the rock at four subsequent positions and is the direct visualization of the type of data obtained by the described workflow. The reliability of the DCR extraction can be judged by the density of pink rocks in panel (a). The revised caption of Figure 4 is marked in blue in the attachment.

*3. I don't think that the few observations made in this experiment (only 5 runs with a wheel-like block) can be generalized to support the conclusions on jump height and mitigation measures dimensioning (p12, l8). Some extra cautions have to be taken before jumping to these conclusions.*

We only present 5 runs in order to facilitate visualization and presentation of the experimental

methodology, as this paper focuses on the 4D reconstruction. The entire data set comprises more than those 5 runs and ongoing analysis corroborates the made claims. We deliberately decided against the inclusion of further data analysis, in order to keep the presented paper in a concise and compact manner. We agree, however, that the presentation of the conclusion seems premature and thus rephrased the statement, pointing towards the ongoing in-depth analysis. As the overestimation of jump heights is a key issue for most simulations programs, we advocate in favour of leaving the statement in, being aware of its implications. The amended sections in the conclusions are marked in blue in the attachment.

*4. It's a Rolls-Royce experiment, involving a heavy logistics (even a helicopter). Such experiments are definitely useful, but the number of runs is naturally low and may be a limitation when studying stochastic processes. Do you think it would be possible to downscale this kind of experiment, keeping the same monitoring techniques?*

Here, we face the major conflict when it comes to real-scale rockfall experiments: Generating a statistically relevant data set without generating enormous costs. The experimental evolution in our group started with small boulders (40-80 kg) which we could carry up the slope. Major criticism has always been that the considered energies are not of practical relevance. Thus, we are currently predominantly aiming for an up-scaling rather than a down-scaling. Now we strive towards a maximum cost efficiency for such large-scale experiments, where the use of a helicopter is not necessarily more expensive than a lengthy installation of a logging ropeway or lending of other heavy machinery such as excavators, cranes, etc. With our approach, we aim for maximal repeatability dealing with rock masses of interest for hazard engineers. We believe, that the amount of deposition points will allow certain conclusions with respect to stochastic processes, but we are fully aware of its limitations.

However, we are also pursuing the ideas of donwscaling the described experiment to laboratory size, with easier control on ground specifications, etc. There, the further development of the videogrammetric reconstruction can be pursued with less expenditure.

In short, yes, we strongly believe that the proposed techniques are applicable in a down-scaled environment, yielding valuable data.

We would like to thank the referee for evaluating our manuscript and strongly believe that the publication was enhanced by the made changes.

On behalf of the authors, yours sincerely, Andrin Caviezel

Please also note the supplement to this comment:
https://www.earth-surf-dynam-discuss.net/esurf-2018-74/esurf-2018-74-AC2-supplement.pdf

**Supplement:**

[revised manuscript text omitted]

---

## Author Response (ED1)

We thank the referee for his positive conception of our submission and his suggestions for improvements.

The manuscript has been amended to incorporate the referee's suggestions in the following way (and can be reviewed in the supplemented file):

- Title: We changed it to "four-dimensional" as the temporal dimension is of significant interest in the shown reconstruction, indeed.

- Abstract: We enlarged the significance statement, as the referee points out correctly,

that a complete understanding of rockfall kinematics is a essential part of general cascading effects in gravitational sediment transport.

- The UAS methodology has been supplemented with addtitional technical information for the interested reader, such that the used settings and workflow is reconstructable.

- We ammended all the minor unclarities (P3,L6; P4,L19; P7,L2;P7,L21) with short additional explanations and additional references.

Additional changes: Figure 1 was ammended, such that colorbar ticks are fully readable and the location of the test site is placed more accurately in the "Switzerland" inset. Any small inconsistencies or unclarities in terms of the DCR work-flow should be phrased more clearly. Figure 4 has also been improved: Caption are ammended, readability and self-explanatory style of figure improved.

On behalf of all of the authors, yours sincerely,

Andrin Caviezel

Please also note the supplement to this comment:
https://www.earth-surf-dynam-discuss.net/esurf-2018-74/esurf-2018-74-AC1-supplement.pdf

[Figure]

We thank for the positive conception of our submission and your suggestions for improvements. We deliberately focused on the experimental methodology and its implementation in order to write a compact publication. An in-detail analysis of the experimental results and their possible implications to constitutive models and calibration routines is on-going work.

Please find below the response on your remaining criticisms:

*1. Rockfall trajectography is one of the most popular topics in the geohazards community, with numerous publications. Very few references are actually given in this paper. I would suggest the authors to introduce shortly how their experiment compares with previous experiments (not only those of the authors' group).*

In order blend our work into the past and ongoing research of rockfall trajectography, we extended the introductory part (Introduction, paragraph 2) by numerous references, focusing especially on published experimental work. As already adked for in RC1, the overall manuscript has been extended with more references to relevant work. Slight rephrasing of the given paragraph, emphasizes the distinction of latest research (Saroglou et al., 2018, Ushiro et al., 2006; Hibert et al., 2017; Salo et al., 2018, our work) on an experimental reconstruction methodology aiming for a full-slope trajectographic reconstruction. The revised introductory paragraph and phrasings are marked in blue in the attachment.

*2. I would appreciate to have a figure with a picture of one of the impact point and the corresponding point cloud after impact. That would help to understand the quality of the data and the type of information available.*

We ammended Figure 4, especially panel (b) and the overal caption such that the quality and type of data of the DCR reconstruction workflow is more easily understandable. Panel (a) depicts a visualization of superimposed reconstructed point clouds of one upper trajectory sector. The pink rock is well distinguishable in most of the image pairs. That is, this figure is a direct visualization of data quality. Inset (b) has been redesigned and labelled such that the extracted point clouds of the rock is emphasised. It shows the residual points of the rock and its centre of mass extraction after colour-filtered extraction of the rock at four subsequent positions and is the direct visualization of the type of data obtained by the described workflow. The reliability of the DCR extraction can be judged by the density of pink rocks in panel (a). The revised caption of Figure 4 is marked in blue in the attachment.

*3. I don't think that the few observations made in this experiment (only 5 runs with a wheel-like block) can be generalized to support the conclusions on jump height and mitigation measures dimensioning (p12, l8). Some extra cautions have to be taken before jumping to these conclusions.*

We only present 5 runs in order to facilitate visualization and presentation of the experimental

methodology, as this paper focuses on the 4D reconstruction. The entire data set comprises more than those 5 runs and ongoing analysis corroborates the made claims. We deliberately decided against the inclusion of further data analysis, in order to keep the presented paper in a concise and compact manner. We agree, however, that the presentation of the conclusion seems premature and thus rephrased the statement, pointing towards the ongoing in-depth analysis. As the overestimation of jump heights is a key issue for most simulations programs, we advocate in favour of leaving the statement in, being aware of its implications. The amended sections in the conclusions are marked in blue in the attachment.

*4. It's a Rolls-Royce experiment, involving a heavy logistics (even a helicopter). Such experiments are definitely useful, but the number of runs is naturally low and may be a limitation when studying stochastic processes. Do you think it would be possible to downscale this kind of experiment, keeping the same monitoring techniques?*

Here, we face the major conflict when it comes to real-scale rockfall experiments: Generating a statistically relevant data set without generating enormous costs. The experimental evolution in our group started with small boulders (40-80 kg) which we could carry up the slope. Major criticism has always been that the considered energies are not of practical relevance. Thus, we are currently predominantly aiming for an up-scaling rather than a down-scaling. Now we strive towards a maximum cost efficiency for such large-scale experiments, where the use of a helicopter is not necessarily more expensive than a lengthy installation of a logging ropeway or lending of other heavy machinery such as excavators, cranes, etc. With our approach, we aim for maximal repeatability dealing with rock masses of interest for hazard engineers. We believe, that the amount of deposition points will allow certain conclusions with respect to stochastic processes, but we are fully aware of its limitations.

However, we are also pursuing the ideas of donwscaling the described experiment to laboratory size, with easier control on ground specifications, etc. There, the further development of the videogrammetric reconstruction can be pursued with less expenditure.

In short, yes, we strongly believe that the proposed techniques are applicable in a down-scaled environment, yielding valuable data.

We would like to thank the referee for evaluating our manuscript and strongly believe that the publication was enhanced by the made changes.

On behalf of the authors, yours sincerely, Andrin Caviezel

Please also note the supplement to this comment:
https://www.earth-surf-dynam-discuss.net/esurf-2018-74/esurf-2018-74-AC2-supplement.pdf

[revised manuscript text omitted]

---

## Author Response (AR2)

**Author's response to Associate Editor Decision on "Reconstruction of three-dimensional rockfall trajectories using remote sensing and rock-based accelerometers and gyroscopes"**

Dear Michael Krautblatter, dear Niels Hovius.

We thank for the in detail treatment of our manuscript and the positive conception of our submission. We are grateful for the meaningful hints to improve final submission and have augmented the manuscript according to the proposoal:

Please find below the response on your remaining suggestions:

> *(i) Introduction: In ESurf, there should also be a more prominent statement highlighting the scientific importance of such data next to the engineering practice. The reach of disc shaped particles with a better rotation of particles has been postulated since the pioneering work of Anders Rapp in the 1960s; many scientific papers have speculated on energy transformation of particles during the impact with the ground and scaled it with particle size, surface material, vegetation etc. and introduced an assumed constant "restitution coefficient". Your data provides very good evidence to reject the existence of such a constant restitution coefficient and you mention it in the conclusion. However, there should be a small paragraph in the Introduction better explaining the scientific debate mentioned above to prepare the wider ESurf readership to be able to follow your assumptions.*

We prominently extended the introductory part (Introduction, paragraph 1) by with a small introductory paragraph (and corresponding references) to position our current research in front of the longstanding – and controversial - discussion on restitution coefficients in rockfall science.  The revised introductory paragraph and phrasings are marked in blue in the attachment.

> *(ii) Starting from the state of the art of "restitution coefficients", it would make sense to provide ratios of ingoing and outgoing kinetic and rotational energies at the impact rather than absolute values to underline the different types of impact. For example in Table 2, I would add a final column at the right with the ratio of E kin b / E kin e as well as E rot b / E rot e such as 0.39/1.03 (first line). These two ratios are the crucial outcome in terms of how much kinetic and rotational energy is proportionally lost (or even gained as in the case of rotational energy) and it is much better readable than the non-normalised absolute values. The ratios are a conciser and better normalised information and you also use them implicitly in the text to explain near-perfect elastical rebounds (p17 l2) and so on. I would also suggest using them in the conclusion/abstract to highlight the differences between the alternative types of ground contacts.*

We very much like this idea and eagerly adopted it to the table. However, the ratios have to be calculated form the respective end velocities of a jump and the consecutive start velocity of its succeeding jump. Since the table rows consist of parameter of interest of individual airborne phases, we added new lines corresponding to the respective impacts between these jumps. The rations differ thus slightly from the upper example. The revised caption of Table 2 is marked in blue in the attachment. The additional information arising from those ratios is also discussed in the corresponding sections in the manuscript (p11, l24 ff,p13, l17ff). We also enriched the conclusion with a concluding remark on the restitution coefficient discussion (p14, l14ff).

> *(iii) In the conclusion, rather than indicating that these experiments can be transferred to other settings (obviously), I suggest to include a sentence what type of information future experiments could add - other types of contact, greater variation of grain sizes, impact on structures...*

We added a paragraph in the conclusion highlighting further experimental ideas treating greater variety of grain sizes, inclusion of mitigation structures etc. (p14 l21ff).

*(iv) In the attached pdf you find ca. 80 smaller remarks on spelling, readability, clarity etc.*

The manuscript has been amended according to the remarks. Thank you very much!

We would like to thank the Associate Editor Michael Krautblatter for his valuable time in evaluating our manuscript and strongly believe that the publication was enhanced by the made changes.

On behalf of the authors, yours sincerely,

Andrin Caviezel